# Abemaciclib for the Treatment of HR+HER2− Metastatic Breast Cancer: An Institutional Experience

**DOI:** 10.3390/cancers16101828

**Published:** 2024-05-10

**Authors:** Erika Matos, Kaja Cankar, Neža Režun, Katja Dejanović, Tanja Ovčariček

**Affiliations:** 1Department of Medical Oncology, Institute of Oncology Ljubljana, Zaloška 2, 1000 Ljubljana, Sloveniatovcaricek@onko-i.si (T.O.); 2Faculty of Medicine Ljubljana, Vrazov Trg 2, 1000 Ljubljana, Slovenia

**Keywords:** CDK4/6 inhibitors, breast cancer, real world

## Abstract

**Simple Summary:**

Abemaciclib in combination with endocrine therapy is a standard first- or later-line of treatment for HR+/HER2− metastatic breast cancer (MBC). Real-world studies provide valuable addition to pivotal trials by including a heterogeneous patient population, particularly subgroups that are often underrepresented (e.g., elderly patients). This study retrospectively collected data on the efficacy and safety of abemaciclib in a real-world patient population. The safety and efficacy of the treatment were consistent with the pivotal studies, including in elderly patients. Comparable survival outcomes were observed between the first two lines of treatment. Additionally, metastatic disease survival was greatly prolonged, exceeding 5 years compared to historical data. The study provides further reassurance that dose reductions do not affect survival. The safety profile of abemaciclib was consistent with previous findings, and no new adverse events (AEs) were identified. Although abemaciclib was well tolerated in elderly patients, the careful monitoring and management of AEs is still warranted.

**Abstract:**

(1) Background: Abemaciclib combined with endocrine therapy is a standard first- or later-line of treatment for HR+/HER2− metastatic breast cancer (MBC). The aim of this retrospective cohort study was to describe the outcomes of patients treated in a real-world setting, with particular focus on elderly patients. (2) Patients and methods: Patients treated with abemaciclib between November 2019 and February 2022 were included in the study. Data were collected from electronic medical records. The primary objective was to determine real-world progression-free survival (rwPFS), and secondary objectives included median overall survival (mOS) and safety. (3) Results: Analysis included 134 patients, with a median follow-up of 42 months. Median age was 62 years, with 29.9% aged 70+ years. A total of 51.5% of patients received abemaciclib in first-line, predominantly with aromatase inhibitor (68.1%). Median rwPFS was 21 in first-line and 20 months in the second-line, with no significant difference between treatment lines (HR 0.96; *p* = 0.88). Patients treated in the third- or later-line had a significantly shorter rwPFS, at 7 months (HR 1.48, *p* = 0.003). mOS was not reached in the first-line setting. For second- and third- or later-lines, mOS was 29 and 19 months, respectively. There was no significant difference in mOS between first- or second-line (HR 1.37, *p* = 0.36). In the 70+ group, median rwPFS was 15 months and mOS was 25 months with no significant difference compared to younger patients (rwPFS HR 1.1; *p* = 0.65; OS HR 1.4; *p* = 0.21). Most common adverse events (AEs) were diarrhoea (68.7%), anaemia (64.9%), and increased serum creatinine (63.4%). Grade 3/4 AEs were reported in 21.6% of patients. Dose reductions occurred in 30.6% of patients and were more frequent in patients 70+ (40%) compared to younger patients (28%); the difference was not significant (*p* = 0.22). At study cut-off, 64.9% of patients discontinued abemaciclib, primarily due to disease progression (73.5%). (4) Conclusions: Our study provides valuable insights into the effectiveness and safety of abemaciclib for the treatment of MBC. We observed comparable outcomes in terms of rwPFS and OS between the first two lines, suggesting consistent effectiveness across treatment lines. In addition, our findings suggest that older age (70+) does not significantly impact the effectiveness and tolerability of abemaciclib, although the careful monitoring and management of AEs are warranted.

## 1. Introduction

Abemaciclib is a cyclin-dependent kinase 4 and 6 (CDK4/6) inhibitor approved for the treatment of hormone receptor positive (HR+), human epidermal growth factor receptor 2 negative (HER2−) metastatic breast cancer (MBC) in combination with endocrine therapy (ET) [1]. Combinations of ET with abemaciclib have led to improved outcomes among patients with HR+/HER2− MBC over ET alone. MONARCH clinical trials demonstrated a clear benefit in prolonging progression-free survival (PFS) regardless of the line of therapy in which abemaciclib was used [1,2]. On the other hand, the statistically significant overall survival (OS) benefit was seen only when abemaciclib was used in second or subsequent lines [3]. As pivotal clinical trials have failed to demonstrate a clear OS benefit of adding abemaciclib to first-line ET, there are a lack of data to help clinicians decide whether abemaciclib is best added to first- or second-line ET. The SONIA trial was designed to test the importance of CDK4/6 inhibitor use in the first-line setting and failed to confirm the benefit of its use in first-line compared to second-line [4,5].

Real-world observational data complement clinical trials as they represent a more heterogeneous patient population. Elderly patients, despite representing a large proportion of breast cancer patients, are often under-represented in clinical trials, putting them at risk of both under- and over-treatment. While older patients were enrolled in MONARCH trials, they may not represent the general older MBC patient population in routine clinical practice. More data are needed on the management of this subgroup [6,7]. 

Few studies have examined the use of abemaciclib in routine clinical practice. To fill the gap of paucity of real-world evidence, this retrospective cohort study was aimed to provide data on the use of abemaciclib in patients with HR+/HER2− MBC, with a special focus on understanding its safety and effectiveness in elderly patients. 

## 2. Patients and Methods

An institutional retrospective analysis of patients treated with abemaciclib between November 2019 and February 2022 for HR+/HER2− MBC was performed. Data on the number of patients treated with abemaciclib were obtained from the Slovenian National Institute of Public Health and the Slovenian Cancer Registry. Baseline demographic and clinical characteristics and treatment patterns of patients were obtained from electronic health records. All patients in this analysis received abemaciclib as systemic therapy in the metastatic setting. Abemaciclib was prescribed with aromatase inhibitor (AI) or fulvestrant in the first, second, or subsequent line of treatment. The index date was defined as the date of initiation of first-line metastatic systemic therapy. Follow-up time was measured from the index date to the patient’s last known structured activity (last visit/death). 

The primary objectives were abemaciclib treatment outcomes defined as median real-world progression-free survival (rwPFS), median overall survival (mOS), and median metastatic overall survival (OSmet). Real-world progression-free survival was defined as the time from the start of abemaciclib treatment to the first documented real-world progression date or death during abemaciclib treatment, mOS from the initiation of abemaciclib treatment, and OSmet from the diagnosis of metastatic disease to death from any cause. Survival analyses were performed for the entire cohort and in subgroups according to age (<70 or ≥70 years) and line of abemaciclib treatment (first, second or third, and subsequent). Due to the retrospective nature of the study, rwPFS was used as a surrogate for the PFS used in the pivotal trials. If patients did not die or had no documented progressive disease, they were censored at the study cut-off date of 21 July 2023. Real-world progressive disease was determined by the treating physician based on radiological, laboratory or clinical assessment. 

The secondary objective was a safety analysis. The real-world adverse events (rwAEs) of interest included diarrhoea, neutropenia, liver enzyme elevation, fatigue, nausea, thrombocytopenia, abdominal pain, and anaemia. Grades for structured laboratory rwAEs were derived based on the National Cancer Institute’s Common Terminology Criteria for Adverse Events (CTCAE), version 5. Dose changes due to rwAEs were recorded. We also examined the association between dose reduction and survival.

### Statistical Methods

Descriptive statistical methods were used to present the patient cohort. Median follow-up was calculated using the reverse Kaplan–Meier method. Estimated survival curves for rwPFS and mOS were generated using the Kaplan–Meier method and compared using the log-rank test. The difference between survival functions was compared using a Cox proportional hazards regression model and expressed as hazard ratio (HR) and 95% confidence interval (95% CI). A *p*-value ≤ 0.05 was considered statistically significant. All statistical analyses were performed using SPSS v.24 (IBM Corp., Armonk, NY, USA).

## 3. Results

### 3.1. Patients and Treatment Characteristics

Between November 2019 and February 2022, a total of 168 patients were treated with abemaciclib in Slovenia. A total of 134 (79.8%) were treated at the Institute of Oncology Ljubljana (IOL)—133 women and one man. Median follow-up was 42 months (95% CI 38.28–45.71). The median age at start of treatment with abemaciclib was 62 years (interquartile range [IQR], 54–71).

In the present cohort, 34 (25.4%) patients had de novo metastatic disease, the majority (70%) of patients were treated with adjuvant chemotherapy (CT) and 92% received adjuvant ET. The median duration of adjuvant ET was 57 months (IQR, 37.2–66.25). 

In our cohort, 51.5% (*n* = 69) of patients received abemaciclib in combination with ET as a first-line treatment of MBC. An AI was the predominant endocrine partner (*n* = 47; 68.1%), while the rest, 31.9% (*n* = 22) of patients received abemaciclib in combination with fulvestrant. Almost half of patients received abemaciclib in later-lines; 32 (23.9%) in second-line and 33 (24.6%) in third- or later-line. The most commonly prescribed type of systemic treatment after progression to abemaciclib was CT (77.7% in the first-line abemaciclib cohort and 75% in the second-line cohort). Capecitabine was the most common. In patients who received abemaciclib as second-line treatment of MBC, the most common first-line treatment was ET (69%): AI (76%), tamoxifen 10%, fulvestrant 14%. The most commonly prescribed first-line CT was anthracycline-based (60%). Other demographic and clinical characteristics of the enrolled patients are shown in Table 1.

### 3.2. Survival Analysis

#### 3.2.1. Real-World PFS

The median rwPFS for patients treated with abemaciclib and ET for entire cohort was 15 months. There was no statistically significant difference when abemaciclib was prescribed in the first- or second-line. However, patients who received abemaciclib in the third- or later-line had a significantly shorter rwPFS, at 7 months (HR 1.48, 95%CI 1.14–1.92, *p* = 0.003). Abemaciclib dose reductions did not affect rwPFS (Figure 1, Table 2). 

#### 3.2.2. Overall Survival from the Initiation of Abemaciclib Treatment

In the entire cohort, the mOS from the initiation of abemaciclib was 29 months. There was no statistically significant difference in mOS between patients who received abemaciclib in the first- or second-line, but expectably significantly shorter times when abemaciclib was initiated in the third- or later-line (HR 1.52, 95% CI 1.13–2.05, *p* = 0.006) (Figure 2, Table 2).

#### 3.2.3. Metastatic Disease OS

The survival from the diagnosis of metastatic disease for the entire cohort was 73 months. There was no difference in OSmet if abemaciclib was prescribed in first- or second-line (Figure 3, Table 2).

#### 3.2.4. Survival Analysis in Elderly Patients

Median rwPFS with abemaciclib and ET in the ≥70 age group was 15 months and 17 months in the <70 age group. Median OS from the initiation of abemaciclib treatment was 25 months (95% CI 20.59–29.40) in the ≥70 age group and 34 months (95% CI 27.44–40.56) in the <70 age group. There was no statistically significant difference in rwPFS and mOS according to age (Figure 4, Table 2).

### 3.3. Safety Analysis

Any grade of rwAEs were reported in 97.8% (*n* = 131) patients. Grade 3 or 4 adverse events were reported in 21.6% (*n* = 29) of patients. Diarrhoea was the most common rwAE and occurred in almost two-thirds of patients. Grade 3 or 4 diarrhoea was more common in elderly patients compared to younger. On the other hand, the frequency of grade 3 or 4 neutropenia was higher in younger patients. Two patients experienced a venous thromboembolic event, one a pulmonary embolism and the other a superficial thrombophlebitis. Table 3 presents most common rwAEs with abemaciclib treatment for all patients and by age groups.

Dose reductions of abemaciclib occurred in 30.6% (*n* = 41) of patients, to 100 mg bid in 27 patients and to 50 mg bid in 14 patients. The most common rwAE leading to dose reduction was diarrhoea (32.2%, *n* = 19). There were more abemaciclib dose reductions in patients aged 70 years or older (40%) compared to younger patients (28%), but the difference was not statistically significant (χ^2^ (1) = 1.98; *p* = 0.22). 

At study cut-off, 64.9% (*n* = 87) of patients discontinued abemaciclib; the most common reasons for discontinuation were disease progression (73.5%, *n* = 64) and adverse events (26.5%, *n* = 23). 

## 4. Discussion

CDK4/6 inhibitors have become a mainstay of treatment for HR+/HER2− MBC, and clinical trials with abemaciclib have demonstrated efficacy across multiple lines of therapy. The present study evaluated real-world treatment patterns and outcomes in patients with MBC treated with abemaciclib at the IOL. To our knowledge, this is the largest European real-world retrospective study with abemaciclib to report an overall survival outcome. In addition, our study reports on survival with abemaciclib in the elderly population, which represents a large proportion of breast cancer patient population but is often under-represented in clinical trials.

The present cohort represents patients prescribed abemaciclib in the early post-approval period, which is often characterised by heterogeneity in drug use, as evidenced by the variable use of abemaciclib across lines of therapy in our cohort [8].

Patients tended to have characteristics indicating a less favourable prognosis compared to patients included in the MONARCH-2 [1] and MONARCH-3 [9] trials. In MONARCH-3, 39.8% of patients had de novo MBC, which is higher than our cohort (25.4%) [9]. In addition, 55.8% and 52.9% of patients in MONARCH-2 and MONARCH-3, respectively, had visceral disease at the start of abemaciclib treatment, compared to 86.6% of patients in our cohort. Bone-only disease was more common in the MONARCH-2 and MONARCH-3 patient population (26.9% and 21.3%, respectively vs. 15.7%) [1,9]. 

Higher level of endocrine resistance in our first-line cohort might be implicated by the greater use of fulvestrant as the first-line endocrine partner (31.9%).

In general, the rwPFS observed in this real-world setting somewhat mirrored what has been observed in randomised clinical trials. Notably, patients treated in the first-line setting had a median rwPFS of 23 months, which is slightly shorter than the median PFS (mPFS) in MONARCH-3 (28.2 months) [2]. On the other hand, patients who received abemaciclib in the second-line had a longer median rwPFS (20 months) compared to mPFS in MONARCH-2 (16.4 months) [3]. Dose reductions (150 mg bid vs. 100 or 50 mg bid) did not affect rwPFS, which is in line with pivotal clinical studies [3,10]. 

Compared to other real-world studies, the Spanish retrospective analysis reported shorter rwPFS in all treatment lines (19.3 months in the first-line setting, 10.3 months in the second-line setting, and 5.5 months in the third-line setting) [11]. In a US real-world study with shorter median follow-up, rwPFS was not reached; the reported overall 12-month rwPFS probability of the entire cohort was 61.7%, which is similar to our cohort (57%) [12]. The UK study reported a better 12-month PFS rates for first-line (81%) and second-line (68.2%), versus 65.5% and 60% in our cohort, respectively [13].

Median OS from the start of abemaciclib treatment in our cohort was not reached at the cut-off date in the first-line setting. It was 66.8 months in MONARCH-3 [10]. In the second-line setting, mOS was shorter in our cohort compared to MONARCH-2, 29.0 vs. 46.7 months, respectively [3,10]. The shorter survival could be explained by the less favourable characteristics of our cohort, such as a higher rate of visceral metastases at the start of abemaciclib treatment, a higher rate of endocrine resistance, and a lower rate of patients with bone disease only. The other real-world studies with abemaciclib did not report OS.

The mOS from the diagnosis of metastatic disease in our cohort was surprisingly long, at 73 months. To our knowledge, no study has reported such a long survival of patients with MBC so far. Moreover, the mOS from the diagnosis of metastatic disease is rarely reported in the studies. The ESME-MBC cohort is a retrospective national cohort, gathering routinely collected real-world data from all consecutive metastatic breast cancer patients who initiated treatment of metastatic disease in the years 2008–2016, in one of the 18 centres in France. The reported mOS from the diagnosis of metastatic disease in 13,590 HR+/HER2− breast cancer patients was 42.9 months, which is significantly shorter than in the present cohort. The main explanation for this difference is probably the use of CDK4/6 inhibitors with only 31% of patients in ESME-MBC cohort receiving CDK4/6 inhibitors in 2016. These data therefore could not yet reflect the impact of CDK4/6 inhibitors on survival [14]. Another explanation for long mOS in our cohort is selection bias; almost a quarter of patients received abemaciclib in the post-approval period as third- or later-line of treatment. The patients were considered fit enough to receive abemaciclib in later lines and did not require a more aggressive treatment, which implicates an indolent disease course. Based on current treatment standards, most patients would receive CDK4/6 inhibitors in a first- or second-line setting at our institution; nevertheless, our results confirm that there is a subgroup of patients with metastatic HR+/HER2− breast cancer who can achieve a durable response with ET alone. 

Clinicians lack sufficient data to determine whether CDK4/6 inhibitors should be added to ET in the first- or second-line. Line of treatment was not found as an independent prognostic factor for survival in our cohort. This result is consistent with the findings of the SONIA trial, which reports no benefit of using CDK4/6 inhibitors in first- versus second-line treatment [4].

Our cohort had a lower rate of diarrhoea (68.7%), compared to the rates reported in MONARCH-2 and -3 (85%) [15], and a similar rate of neutropenia as reported in the pivotal clinical trials (MONARCH-2 46%; MONARCH-3 43.7%) [1,9]. The rate of abemaciclib dose reductions was higher in pivotal trials (MONARCH-2 and -3, at 42.9% and 46.5%, respectively) compared to our cohort (30%). These results suggest that treatment with abemaciclib is well tolerated in routine clinical practice. 

There is a lack of evidence regarding the treatment of elderly cancer patients with abemaciclib due to their under-representation in clinical trials. However, several studies have shown that the appropriate management of abemaciclib treatment in geriatric cancer patients may lead to prognostic improvements. Focusing on age, the median age in our cohort was 62 years, which is similar to the median age of patients enrolled in MONARCH-2 and MONARCH-3 (59 and 63 years, respectively) [1,9]. In our cohort, one-third of patients (29.9%) were elderly, ≥70 years of age. There was no significant difference in the abemaciclib treatment effectiveness between patient groups according to age (<70 vs. ≥70 years of age), which is consistent with the results of MONARCH-2 and MONARCH-3 [16] and indicates a similar effectiveness of abemaciclib in the elderly patient population. In MONARCH trials, safety analysis did not find a higher incidence of AEs in the elderly patient population, which was also the case in our cohort [16]. In our elderly patient population, there were lower incidences of reported rwAEs compared to pivotal trials, which could be the consequence of a less strict reporting of AEs in routine clinical practice. Dose reductions were more common in elderly patient population, although the difference was not statistically significant. This was also observed in the MONARCH trials [16]. 

The limitations of observational retrospective studies, such as the retrospective collection of data and under-reporting of AEs in patients’ medical records and patient population heterogeneity (age, line of treatment, and ET) should be taken into consideration when interpreting the findings of this study. In addition, the patient population is small due to abemaciclib being a relatively new drug and there are other available drugs in the same therapeutic class. 

## 5. Conclusions

In conclusion, the results of the present study support the results from registration studies in terms of abemaciclib safety and effectiveness in HR+/HER2− MBC patients, including in elderly patients. We observed comparable survival outcomes between the first two lines of treatment. Notably, patients in third- or later-lines had shorter survival, highlighting the importance of exploring alternative therapeutic strategies in more advanced disease. Our study provides further reassurance that dose reductions do not affect survival rates. Safety profile was consistent with previous findings with no new AEs identified. Abemaciclib was well tolerated in elderly patients, but the careful monitoring and management of AEs are still warranted.

## Figures and Tables

**Figure 1 cancers-16-01828-f001:**
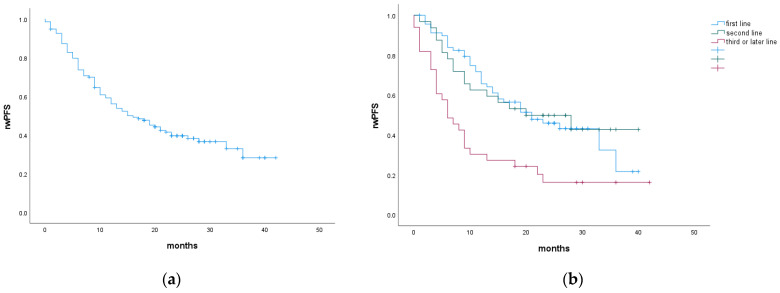
rwPFS for the entire cohort (**a**) and according to abemaciclib treatment line (**b**).

**Figure 2 cancers-16-01828-f002:**
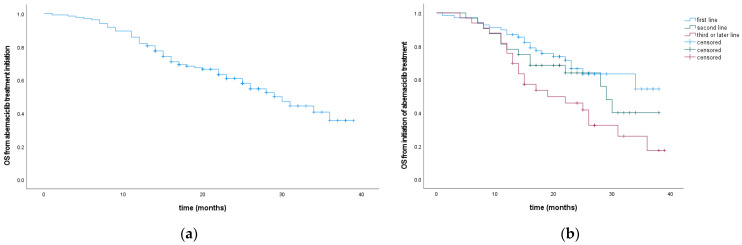
OS from the introduction of abemaciclib for the entire cohort (**a**) and according to abemaciclib treatment line (**b**).

**Figure 3 cancers-16-01828-f003:**
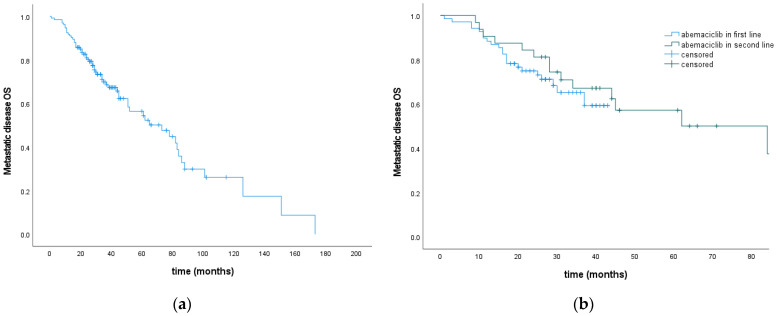
metOS for the entire cohort (**a**) and according to abemaciclib treatment line (**b**).

**Figure 4 cancers-16-01828-f004:**
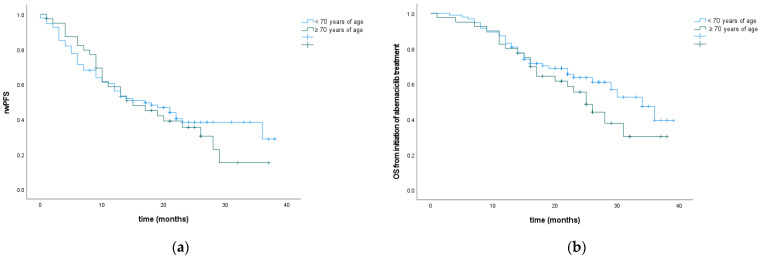
rwPFS (**a**) and OS (**b**) according to patient age.

**Table 1 cancers-16-01828-t001:** Patients’ characteristics.

Characteristic	
Age at abemaciclib initiation	
Median (IQR)	62 (54–71)
Age at abemaciclib initiation, N (%)	
Age < 70	94 (70.1)
Age ≥ 70	40 (29.9)
Sex, N (%)	
Female	133 (99.3)
Male	1 (0.7)
Metastatic status at presentation, N (%)	
De novo	34 (25.4)
Recurrent	100 (74.6)
Metastatic sites at abemaciclib start, N (%)	
Bone only disease	21 (15.7)
Visceral disease	116 (86.6)
Liver metastases	59 (44.0)
CNS	8 (5.9)

CNS = central nervous system, N = number.

**Table 2 cancers-16-01828-t002:** Survival analysis results.

Survival	Months (95% CI)	HR (95% CI); *p*
rwPFS		
Entire cohort	15 (9.52–20.48)	
According to treatment line		
First-line	21 (15.12–26.88)	
Second-line	20 (6.38–33.60)	0.96 (0.54–1.69); ns
Third-line or later	7 (4.19–9.81)	1.48 (1.14–1.92); 0.003
According to dose		
150 mg bid	16 (3.71–8.74)	
100 mg or 50 mg bid	15 (4.14–6.89)	1.06 (0.67–1.69); ns
According to age		
<70 years	17 (10.10–23.89)	
≥70 years	15 (6.99–23.01)	1.10 (0.70–1.76); ns
mOS		
Entire cohort	29 (24.15–33.84)	
According to treatment line		
First-line	NR	
Second-line	29 (26.09–39.90)	1.37 (0.69–2.69); 0.36
Third-line or later	19 (7.59–30.49)	1.52 (1.13–2.05); 0.006
According to dose		
150 mg bid	30 (23.01–36.95)	
100 mg or 50 mg bid	28 (20.36–35.64)	1.16 (0.68–1.99); ns
According to age		
<70 years	34 (27.44–40.56)	
≥70 years	25 (20.59–29.40)	1.4 (0.82–2.39); ns
OSmet		
Entire cohort	73 (45.46–100.35)	
According to treatment line		
First-line	NR	
Second-line	84 (31.27–136.73)	0.75 (0.35–1.60); ns

mOS = median overall survival, rwPFS = median real-world progression-free survival, OSmet = median metastatic disease overall survival, NR = not reached, bid = twice a day, ns = not significant.

**Table 3 cancers-16-01828-t003:** Adverse events with abemaciclib treatment for all patients and by age groups.

AE	All Patients N (%) *n* = 134			<70 N (%) *n* = 94			≥70 N (%) *n* = 40		
	Any Grade	G2	G3/4	Any Grade	G2	G3/4	Any Grade	G2	G3/4
Diarrhoea	92 (68.7)	27 (20.1)	7 (5.2)	68 (72.3)	20 (21.3)	4 (4.3)	24 (60)	7 (17.5)	3 (7.5)
Neutropenia	61 (45.5)	39 (29.1)	17 (12.9)	48 (51.1)	30 (31.9)	15 (16.0)	13 (32.5)	9 (22.5)	2 (5.0)
Anaemia	87 (64.9)	21 (15.7)	1 (0.7)	62 (66.0)	14 (14.9)	0	25 (62.5)	7 (17.5)	1 (2.5)
Nausea	40 (29.9)	13 (9.7)	1 (0.7)	28 (29.8)	9 (9.6)	0	12 (30.0)	4 (10.0)	1 (2.5)
Fatigue	52 (38.8)	12 (9.0)	3 (2.2)	35 (37.2)	8 (8.5)	1 (1.1)	17 (42.5)	4 (10.0)	2 (5.0)
Abdominal pain	30 (22.4)	3 (2.2)	0	22 (23.4)	2 (2.1)	0	8 (20.0)	1 (2.5)	0
Thrombocytopenia	25 (18.7)	1 (0.7)	0	15 (16.0)	0	0	10 (25.0)	1 (2.5)	0

## Data Availability

The data presented in this study are available on request from the corresponding author due to institutional sharing restrictions. A data transfer agreement should be obtained prior to sharing.

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
