# Peer review of "Abemaciclib for the Treatment of HR+HER2− Metastatic Breast Cancer: An Institutional Experience"

_cancers, 2024, doi:10.3390/cancers16101828_

Round 1
Reviewer 1 Report
Comments and Suggestions for Authors
It is a very good paper; the results are in line with legitimate expectation, sopported by other paper on the same subject. Very clear data tables and graphs.
Author Response
Dear Reviewer,
Thank you for the detailed reading of our manuscript. We are glad that you liked the manuscript it its present form.
We have taken other reviewers suggestions into account and improved the manuscript accordingly. We have highlighted the corrections in yellow in the manuscript.
Kind regards,
Erika Matos, MD, PhD
Reviewer 2 Report
Comments and Suggestions for Authors
This retrospective study entitled: "Abemaciclib for the treatment of HR+HER2- metastatic breast cancer: an institutional experience" performed by Matos E et al. shows the real-world application of abemaciclib in treating HR+/HER2- metastatic breast cancer (MBC), particularly emphasizing its safety and efficacy in elderly patients.
The study, conducted between November 2019 and February 2022, included 168 patients from Slovenia.
Key findings indicate a median real-world progression-free survival (rwPFS) of 15 months across all cohorts, with no significant difference in early treatment lines but shorter rwPFS at later stages.
My congratulations to the authors of the study, which enriches the existing literature by confirming the clinical trial outcomes of abemaciclib in a routine clinical setting.
I just have some minor comments:
- Why did you use the inverse KM method to calculate the median follow-up? Can you give an explaination about that?
- It is certainly true that abemaciclib + ET prolongs survival in ER+/HER2- metastatic breast cancer patients but it is very importanto to add that systemic therapy + loco-regional treatment can represent ad important association of treatment in a selected population of oligometastatic breast cancer patients (please cite: PMID: 36551722)
- The main limitation of this this is the small patient population and it should be acknowledged.
Author Response
Dear Reviewer,
Thank you for the detailed reading and all the useful comments. We have taken your suggestions into account and improved the manuscript accordingly. We have highlighted the corrections in yellow in the manuscript and some of them are further explained in the comments below.
Kind regards,
Erika Matos, MD, PhD
- Why did you use the inverse KM method to calculate the median follow-up? Can you give an explanation about that?
Thank you for your comment. The method was advised by a professional institutional statistician, as a method that is accepted for calculating follow-up time in oncology trials. The more appropriate term to describe the method is "reverse Kaplan-Meier method". This has been corrected in the manuscript.
- It is certainly true that abemaciclib + ET prolongs survival in ER+/HER2- metastatic breast cancer patients but it is very important to add that systemic therapy + loco-regional treatment can represent an important association of treatment in a selected population of oligometastatic breast cancer patients (please cite: PMID: 36551722)
Thank you for your comment. We have read the reference you provided which showed no significant survival benefit of adding LRT to systemic treatment in primary oligometastatic breast cancer. The study is interesting, but we believe that the focus of the study is not consistent with this analysis.
- The main limitation of this this is the small patient population and it should be acknowledged.
We agree with reviewers comment. As this is a relatively new drug and there are others available from the same therapeutic class, the number of patients is low. The comment was added in the manuscript (line 293).
Reviewer 3 Report
Comments and Suggestions for Authors
This manuscript reports the real-world data of abemaciclib treatment in combination with endocrine therapy in breast cancer patients of the Slovenian National Institute of Public Health and the Slovenian Cancer Registry. The authors compared the efficacy of treatment according to the age of the patients and line of abemaciclib treatment. The authors compared their findings with that was observed in the MONARCH clinical trials and other real-world studies and reinforce the importance of their observations in older patient group for clinical care.
Minor comments
1- Results: in lines 112-114: the sentence “This section…” could be eliminated.
2- Conclusion: in line 305: “patents” it should say “patients”.
Author Response
Dear Reviewer,
Thank you for the detailed reading and all the useful comments. We have taken your suggestions into account and improved the manuscript accordingly. We have highlighted the corrections in yellow in the manuscript and some of them are further explained in the comments below.
Kind regards,
Erika Matos, MD, PhD
- Results: in lines 112-114: the sentence “This section…” could be eliminated.
Thank you for your comment. The left in the text was overlooked.
- Conclusion: in line 305: “patents” it should say “patients”.
Thank you for your detailed reading, the typo was corrected.